# Preventing Bacterial Translocation in Patients with Leaky Gut Syndrome: Nutrition and Pharmacological Treatment Options

**DOI:** 10.3390/ijms23063204

**Published:** 2022-03-16

**Authors:** Agata Twardowska, Adam Makaro, Agata Binienda, Jakub Fichna, Maciej Salaga

**Affiliations:** Department of Biochemistry, Faculty of Medicine, Medical University of Lodz, 90-419 Lodz, Poland; agata.twardowska@stud.umed.lodz.pl (A.T.); adammakaro@gmail.com (A.M.); agata.binienda@gmail.com (A.B.); jakub.fichna@umed.lodz.pl (J.F.)

**Keywords:** leaky gut syndrome, bacterial translocation, nutrients, probiotics

## Abstract

Leaky gut syndrome is a medical condition characterized by intestinal hyperpermeability. Since the intestinal barrier is one of the essential components maintaining homeostasis along the gastrointestinal tract, loss of its integrity due to changes in bacterial composition, decreased expression levels of tight junction proteins, and increased concentration of pro-inflammatory cytokines may lead to intestinal hyperpermeability followed by the development of gastrointestinal and non-gastrointestinal diseases. Translocation of microorganisms and their toxic metabolites beyond the gastrointestinal tract is one of the fallouts of the leaky gut syndrome. The presence of intestinal bacteria in sterile tissues and distant organs may cause damage due to chronic inflammation and progression of disorders, including inflammatory bowel diseases, liver cirrhosis, and acute pancreatitis. Currently, there are no medical guidelines for the treatment or prevention of bacterial translocation in patients with the leaky gut syndrome; however, several studies suggest that dietary intervention can improve barrier function and restrict bacteria invasion. This review contains current literature data concerning the influence of diet, dietary supplements, probiotics, and drugs on intestinal permeability and bacterial translocation.

## 1. Introduction

Maintaining the proper intestinal barrier permeability is a critical condition to avoid gastrointestinal (GI) tract dysfunction. A large-scale multinational study suggests that more than 40% of people worldwide have functional GI disorders, which affect quality of life and health care use [1]. Leaky gut syndrome (LGS), which is characterized by intestinal hyperpermeability, encompasses a range of systemic disorders [2]. Strong evidence suggests a correlation between LGS and both GI and non-GI diseases, such as inflammatory bowel disease (IBD) [3,4], irritable bowel syndrome (IBS) [5], colorectal cancer (CRC) [6], allergies [7], Huntington’s disease [8], and Parkinson disease [9]. The main symptoms of LGS, including acute or chronic diarrhea or constipation, bloating, fatigue, and headaches, are mostly related to the underlying cause of this disorder, which may be intestinal hyperpermeability [10]. Diarrhea in LGS is primarily due to intestinal epithelium disruption, which may be caused by tight junctions (TJs) dysfunction and/or microbial or viral pathogens. Interestingly, one of the methods used to measure intestinal permeability is the assessment of bacterial biomarkers like circulating endotoxin (e.g., lipopolysaccharide, LPS) or soluble CD14 [11]. Systemic endotoxemia was found in patients with IBD (88% of ulcerative colitis (UC) patients and 94% of Crohn’s disease (CD) patients) and correlated positively with the anatomic extent and clinical activity of UC [12]. In another study, endotoxemia was present in 28% of the patients with UC and in 48% of the patients with CD [13]. Furthermore, patients with IBS have increased expression of toll-like receptor (TLR) 4, which recognizes bacterial LPS, and TLR5, which recognizes bacterial protein, flagellin [14]. Schoepfer et al. [15] showed that anti-flagellin antibodies were significantly more frequent in serum of IBS patients than in healthy controls. Moreover, lipopolysaccharide (LPS) enhances the motility of CRC cells by increasing the release of vascular endothelial growth factor C [16]. 

The intestinal barrier involves several physiological and biochemical elements, such as mucus, immune cells, epithelial cells with TJs, and microbiota. The cornerstone of increased intestinal permeability is a dysfunction of TJ in the epithelium. Disturbance of TJs may be caused by several factors, such as dysbiosis, hypoperfusion, infections, and toxins [11,17]. Gut dysbiosis is a persistent imbalance of the gut microbial community through—among others—bacterial translocation (BT). Many factors may contribute to dysbiosis: increased intake of protein, sugar, or food additives; alcohol; drugs; poor hygiene; stress; or anxiety. Several studies have shown that intestinal dysbiosis is responsible for epithelial barrier dysfunction [18]. *E. coli*, *Klebsiella*, *Proteus*, *Enterobacter*, *Shigella*, *Salmonella*, and *Serratia* are the bacterial species most often related to BT [19]. In this review, we present several factors that influence intestinal permeability through BT and subsequently enhance LGS and other diseases pathophysiology (Figure 1).

## 2. Literature Search

The following databases were searched: PubMed, Embase (OVID version), and Google Scholar. The search query consisted of the combination of the following keywords: “Leaky gut syndrome”, “Bacterial translocation”, “Treatment”, “Probiotic”, “Western diet”, “Dietary lipids”, “Vitamin”, “Bacterial translocation prophylaxis”, and “Dietary supplements”. Results were limited to relevant articles published in English. There were no time restrictions for the articles cited in the manuscript. The first search was performed on 10 March 2021, updated on 10 July 2021, with a final revision on 10 September 2021. Initially, we found 350 articles, of which 182 were included in the review. References of all included studies were reviewed for more eligible studies. Each article was reviewed independently by two (AT, MS) researchers for inclusion according to prior established inclusion and exclusion criteria. Disagreements on article selection were resolved through discussion until consensus was reached or resolved by discussion with AB and AM. Conference abstracts were excluded. Articles were excluded in case of non-English language, inaccessibility of the full text, and commentaries. Extracted data included study population and demographics, details of interventions and controls, study methodology, and information to assess bias. Data extraction was conducted independently by two authors, and discrepancies were resolved through discussion with other co-authors.

## 3. Intestinal Microbiome and Bacterial Translocation

In humans, the lumen of the GI tract is inhabited by an enormous number of bacteria, archea, fungi, and protozoa, which altogether constitute the gut microbiome. The quantity of microorganisms and their species diversity varies depending on the region of the GI tract [20]. In the stomach, gastric acid forms an unfavorable environment for bacterial growth; thus, the number of microorganisms and their variety are highly limited. The microbial density of the stomach hovers around 10^1^–10^3^ CFU/g and mainly consists of *Helicobater*, *Bacterioides*, and *Lactobactillus* genera. Similarly, a low pH level restricts bacteria growth in the duodenum [21]. In contrast to the upper GI tract, the number of microorganisms inhabiting intestines rapidly increases up to even 10^8^ CFU/g in the colon. Obligate anaerobes, such as *Bacterioides*, *Clostriudium*, *Bifidobacterium*, *Fusobacterium,* and facultative anaerobes as well as aerobes, including *Enterobacteriaceae*, *Lactobacillus,* and *Enterococcus*, are the main genera characteristic for intestines. Underway of co-evolution, humans have developed the ability to tolerate commensal bacteria. The interplay between the host and gut microbiome has resulted in a number of biochemical mechanisms, which maintain intestinal homeostasis and reduce the probability of infection [22]. Studies on germ-free (GF) mice showed that the gut microbiome is an essential component in preserving intestinal structure and function [23]. End products of bacterial fermentation of indigestible polysaccharides, such as short-chain fatty acids (SCFAs), are not only a principal energy source of colonocytes but also influence several biological processes, including cellular differentiation and apoptosis of the colonic epithelial cells [24]. Moreover, the intestinal microbiome is directly involved in defense against pathogenic factors. Certain bacterial strains can influence the concentration of both proinflammatory and anti-inflammatory cytokines and modulate the activation and function of immune cells. Additionally, by competition for nourishment and adherence spots, commensal bacteria present in the lumen of the GI tract restrict the growth of potentially pathogenic strains; thereby, they prevent the development of infection. Novel lines of evidence also suggest that bacterial structural components and bacterial metabolites influence intestinal permeability [25]. For example, *Akkermansia muciniphilia*—an intestinal symbiont that is currently a very popular topic of studies among researchers—is evidenced to have anti-inflammatory properties and improves intestinal barrier integrity. *Akkermansia muciniphila*-derived extracellular vesicles administered to high-fat diet (HFD) induced diabetic mice enhanced TJ expression and reduced intestinal permeability in Caco-2 cells [26]. Furthermore, *Akkermansia muciniphila* upregulates *Claudin 3* and *Occludin* through the activation of TLR2 signaling and is considered a mucin-degrading species which normalizes inner mucus layer thickness [27]. Additionally, *Lactobacillus plantarum* has been evidenced to exhibit many beneficial effects on gut barrier integrity. In enterotoxigenic *E. coli*-treated porcine intestinal IPEC-J2 cells, commonly used in research concerning intestinal epithelium function, probiotic *Lactobacillus plantarum* restored claudin-1, occludin, and ZO-1 expression and downregulated TNF-α, IL-6, and IL-8 expression. In piglets, *Lactobacillus plantarum* probiotic increased the abundance of butyrate-producing bacteria, which correlated with the increase in SCFAs level in stool [28]. Although the presence of the gut microbiome brings many benefits to the host, their translocation to non-GI tissues can possibly result in many serious health problems.

In normal conditions, the gut microbiome is separated from the internal environment by an intestinal barrier. However, in some circumstances, dysfunction of gut barrier integrity may enable noxious microorganisms, their antigens, and their toxic metabolites to cross the intestinal barrier and firstly enter mesenteric lymph nodes (MLNs) and then by systemic circulation enter normally sterile host tissues and organs, including liver, lungs, or brain. This phenomenon is defined as BT [29]. BT has been associated with GI-related diseases such as CD, UC, Celiac disease, cirrhosis, and clinical conditions among burn injury, obstructive jaundice, acute pancreatitis, abdominal surgery, bowel transplant, malignancy, and hemorrhagic shock [30,31,32,33]. In most cases, BT occurs as a result of the disease or condition associated with inflammation, oxidative stress, injury, dysbiosis, and impairment of the intestinal barrier. The permeability of the intestinal barrier is influenced by many factors, and the most important seems to be the interactions between the gut microbiome and the host. For example, it was shown that *Enterococcus gallinarum* is involved in the LGS-like phenotype with enhanced BT due to downregulation of TJ-related molecules, such as *Claudin 3* and *Occludin* [18,34]. Two mechanisms of BT and translocation of bacterial molecules may be distinguished: the paracellular pathway and transcellular pathway, which may occur separately or in combination. The paracellular pathway, which occurs more commonly, involves the disruption of TJ, which, in consequence, leads to the damage to the cytoskeleton, its actin filaments, and microtubules and facilitates BT. In contrast, the transcellular pathway is under the control of specific enterocyte channels and membrane pumps. Moreover, the transcellular pathway uses a primary and a secondary active transport across the intestinal epithelial cell through both the apical and the basolateral membranes [35]. Recent reports show that some bacteria, such as *E. coli* and *Proteus mirabilis*, ensure transcellular passage through enterocytes by pinocytosis [36].

Human studies have linked disruptions of intestinal bacteria quantity and diversity to the development of several human diseases in which intestinal hyperpermeability has been detected, including IBD, obesity, diabetes, or ALD; however, the exact mechanisms of these relationships largely remain unknown. In cirrhotic rats with BT or spontaneous bacterial peritonitis (SBP), the total intestinal aerobic bacterial count is significantly higher compared to animals without BT or SBP, suggesting that bacterial overgrowth may directly favor BT [37]. In addition, a correlation between small intestinal bacteria overgrowth (SIBO) and BT was found in cirrhotic patients by Jun et al. [38]. The important role of intestinal microbiota in the regulation of intestinal permeability was highlighted by Madsen et al. [39]. According to the outcomes of their experiment, IL-10 gene-deficient mice developed increased colonic and ileal permeability with simultaneous enhancement in proinflammatory mucosal cytokines (TNF-α and IFN-γ) levels. These changes occurred before the onset of inflammation and were absent in GF mice, suggesting that the relationship between microbiota and the host immune system plays a key role in the onset of increased intestinal permeability [39]. Contraindicatory, in a mouse model of liver disease, BT occurs simultaneously with the decrease in TJs expression and intestinal hyperpermeability and independently to bacteria overgrowth [40]. 

## 4. The Role of Diet in Preventing Bacterial Translocation

The process of translocation of whole bacteria or its molecules is highly dependent on the intestinal wall integrity, which is essentially maintained by TJ proteins. Numerous studies suggest that several dietary compounds have profound influences on its expression and function. For example, the alterations in TJ proteins are responsible for intestinal barrier impairment caused by fructose. It was previously concluded that excessive amounts of fructose in the form of monosaccharide or as a part of sucrose should be limited in a diet of LGS patients [41]. Recently, it has been suggested that chronically increased levels of glucose in the blood may also lead to the disruption of TJ proteins [42]. On the other hand, it is recommended to increase intake of dietary fiber, which is converted by gut microbiota to SCFAs, which enhance intestinal wall integrity. The exact mechanisms of action of SCFAs on intestinal barrier physiology are not fully explained. However, it is known that SCFAs are essential for colon epithelial cells nutrition and protect against TJ alterations [41]. The impact of diet is not limited to the direct influence of its contents on TJ proteins. Current literature provides large evidence that modern habits, including the sedentary lifestyle and western-type diet (WD), affect the GI tract and may contribute to BT. The WD is characterized by high amounts of processed foods with increased caloric content. It is high in saturated fats, proteins, simple sugars, salt, and decreased intake of plant-derived foods [43]. Moreover, it is well established that a diet rich in fats interacts with intestinal microflora composition leading to dysbiosis. 

### 4.1. The Amount and Composition of Fats as Key Dietary Factors Responsible for Endotoxemia and BT: Animal Studies

Fats belong to the basic nutrients that are essential for homeostasis. However, many observations from animal models of diet-induced obesity showed that high intake of fats leads to alterations in gut microbiota profile. Such diets tend to reduce the abundance of *Bifidobacteria*, which was particularly emphasized by Cani et al. [44,45]. They observed that mice fed an HFD for 14 weeks had extremely decreased levels of *Bifidobacteria* and other mouse-specific strains. Increased doses of fat resulted in significant endotoxemia. There is a negative correlation between the plasma endotoxin level and *Bifidobacterium*, which was not observed for other genera. Additionally, the concentration of plasma LPS positively correlates with proinflammatory cytokines, percentage of visceral fat tissue, and body weight gain. Other studies showed that dysbiosis triggered by high doses of fat is characterized by elevated *Firmicutes*/*Bacteroidetes* ratio [46,47,48,49,50,51]. Moreover, these changes resulted directly from diet composition and were not caused by obesity. It was shown by comparing the effects of HFD in wild-type and resistin-like molecule β (RELMβ) knockout (KO) mice. This adipocytokine shares homology with resistin and plays a role in many mechanisms, including glucose and lipid metabolism. Both groups represented similar alterations, but the latter ones were obesity resistant [48]. In another study, both the high-fat and high-sucrose diets promoted the switch from *Bacteroidetes* to *Firmicutes* [49]. The occurrence of endotoxemia resulting from high-fat consumption was also observed by Hamilton et al. [50]. The authors noted that LPS-binding protein (LBP) concentration was significantly increased after 3 and 6 weeks of the dietary intervention. The measurement of para- and transcellular transport with Ussing chambers revealed excessive flux across intestinal walls. The paracellular hyperpermeability was detected through both small and large intestine, but transcellular pathways were affected only in cecal and colonic samples. Moreover, this study showed numerous HFD-induced alterations in the morphology of both the ileal and cecal tissues.

According to several studies, intestinal alkaline phosphatase (IAP) is essential in maintaining proper intestinal barrier function. IAP is an isoform of alkaline phosphatase produced in the small bowel, where it is primarily responsible for the detoxification of the LPS in the luminal layer of mucosa. Moreover, pretreatment with IAP prevented the translocation of whole bacteria to the bloodstream, liver, and lung in the mouse model of peritonitis [52]. Of note, the levels of IAP are dependent on the fats content in the diet. The HFD resulted in decreased IAP and increased plasma LPS in obese rats [53]. In another study, transgenic mice with upregulated expression of IAP were more resistant to increased levels of plasma LPS during the trial of a high-cholesterol diet [54]. It was also observed that curcumin and high doses of n-3 polyunsaturated fatty acids (PUFAs) increase IAP levels and restore intestinal barrier function leading to decreased levels of metabolic endotoxemia [55,56]. These facts ought to be taken into consideration while designing therapeutic diets, which prevent leaky-gut consequences.

It was shown that the impact of HFD on the microflora profile is greatly dependent on the composition of fats. Diets rich in saturated triglycerides lead to dysbiosis and endotoxemia, which are less likely to occur with diets containing more unsaturated fats. This phenomenon is associated with the overactivity of pathways dependent on TLRs, which facilitate LPS and bacteria transport. Current literature suggests that TLR4 signaling is activated by saturated fats and muted by n-3 PUFAs [51,57,58]. It is worth mentioning that the WD is characterized by excessive amounts of saturated triglycerides found in palm oil, butter, and lard. In contrast, the Mediterranean diet (MD) is rich in monounsaturated fatty acids (MUFAs) and PUFAs, including n-3 PUFAs, derived from fish and olive oil. It also contains high amounts of dietary fiber and plant-derived antioxidative compounds, including flavonoids and non-flavonoids. There is large evidence suggesting the beneficial effects of the Mediterranean diet in maintaining a proper microbiota profile [59].

The beneficial properties of olive oil were examined in more detail by Hidalgo et al. [60]. They observed that both diets rich in high-purity virgin olive oil or predominantly refined form of olive oil are protective for gut commensal microbiota, which was essentially impaired by a diet rich in butter. Among these three HFDs, only the one based on virgin olive oil was not significantly different from control mice fed with a standard laboratory diet. This unique form of olive oil is obtained by mechanical extraction from olives without any chemical procedure. Such a way of processing allows maintaining high amounts of antimicrobial and antioxidant phenolic compounds, which are absent in refined forms. In another study, the phenolic compounds obtained from green tea were shown to prevent the mouse intestine from alterations in TJ proteins and dysbiosis caused by HFD [61]. 

### 4.2. The Clinical Role of Diet-Induced Changes in Microbiota Profile

Human studies show that the WD dietary patterns significantly facilitate metabolic endotoxemia, which may occur even immediately after a high-fat meal [62,63]. On the other hand, it is suggested that adherence to MD habits is beneficial for microbiota profile. It was shown that the MD leads to a decrease in *Firmicutes/Bacteroidetes* ratio and increases the number of *Bifidobacteria* or *Lactobacillus*. High amounts of plant-derived nutrients, including dietary fiber and PUFAs, are the essential factors contributing to this phenomenon [64]. Simoes et al. [65] evaluated the impacts of MUFA, n-3 PUFA, n-6 PUFA, soluble fiber, and total energy intake on microflora profile in the retrospective monozygotic twin study. They showed that higher consumption of n-3 PUFA increases the levels of *Lactobacillus* and *Bifidobacteria* in stool samples. The research has not shown other dietary factors promoting *Lactobacillus* growth. Another study demonstrated that the postprandial serum endotoxin levels are significantly lower after consumption of a meal rich in n-3 PUFAs [66]. In summary, dietary habits characteristic for MD should be considered in the prevention of the diseases in which a significant increase in intestinal permeability is noted, e.g., LGS.

Current literature provides large evidence that LGS predispose to the development and progression of liver disorders. It has been shown in patients suffering from non-alcoholic fatty liver disease (NAFLD), including non-alcoholic fatty liver (NAFL) and non-alcoholic hepatosteatosis (NASH). Human studies showed that NAFLD is strongly associated with differences in the gut microflora composition as well as higher levels of plasma LPS, which may be triggered by WD dietary patterns [67]. Moreover, there is a wealth of evidence from animal models of NAFLD, in which endotoxemia induced by high-fat, high-sucrose, or high-fructose diets resulted in the increase in fat accumulation in hepatocytes [68,69,70,71,72,73]. Bergheim et al. [70] particularly highlighted the axis of the high-fructose diet, leaky gut, and the NAFLD development in mice. The antibiotic treatment with polymyxin protected from endotoxemia and resulted in the decrease in hepatic lipid and triglycerides accumulation. On the other hand, it is urgently needed to evaluate the clinical role of dietary interventions in NAFLD. A recent observational study did not show any changes in intestinal permeability of patients who underwent 16 week periods of MD, wash-out diet, and low-fat diet [74]. 

Of note, the LGS is observed in the disorders from the spectrum of alcoholic liver diseases (ALD). It is well established that alcohol and its metabolites are independent factors leading to intestinal wall disruption through multiple mechanisms, including dysbiosis, alterations in TJ proteins, and activation of TLR4 signaling. As a result, endotoxemia plays a key role in the pathogenesis of alcoholic liver injury [75]. For this reason, the diet preventing LGS should be completely alcohol free. On the other side, the harmful effects of alcohol consumption may be alleviated by different compositions of dietary fats. It was observed that increased content of saturated long-chain fatty acids (LCFAs) prevents alcohol-induced dysbiosis and intestinal wall impairment in mice [76,77]. Future clinical studies on the role of specific dietary fats in LGS prevention are therefore needed.

## 5. Probiotics

Probiotics modulate gut microbiota, which results in intestinal wall healing and prevents BT and endotoxemia. In particular, this has been shown in numerous studies concerning *Bifidobacteria* and *Lactobacillus*. Of note, the probiotic therapies consisting of these strains reflect the effects of MD on the microflora. Bagarolli et al. [78] showed that simultaneous supplementation of *Bifidobacteria* and *Lactobacillus* attenuated TLR4 activation in HF-fed mice. Moreover, probiotics significantly reduced inflammation and decreased LPS serum level, which was in line with previous observations from the human colonic microbiota model [79]. On the other hand, the pretreatment with *Bifidobacteria* before induction of hemorrhagic shock in rats resulted in decreased endotoxemia and BT to mesenteric lymph tissue (MLT) [80]. Moreover, *Bifidobacterium animalis* ssp. *lactis 420* attenuated *E. coli* BT triggered by ketogenic and HFDs [81,82]. The positive impact of *Bifidobacteria* was also shown by prebiotic oligofructose, which selectively restores this strain. The intervention protected mice from HFD-induced endotoxemia [44]. 

The supplementation with *Lactobacillus salivarius* reduced endotoxemia and BT to MLT caused by hyperglycemia induced by streptozocin [83]. The positive influence of *Lactobacillus* on TJ proteins was shown in numerous studies focused on liver disorders, which highlight the multifactorial influence of this strain on epithelial function [84]. The administration of *Lactobacillus rhamnosus GG* (LGGs) maintained the proper expression of occludin and claudin-1 and prevented increased plasma LPS concentration during the high-fructose treatment [85]. Similar properties of the strain were observed in mice exposed to alcohol, in which the LGGs supplementation protected against the downregulation of claudin-1 and ZO-1. As a result, alcohol-induced translocation of E. coli protein was significantly reduced [86].

The positive influence of probiotics in preventing LGS was particularly observed in clinical studies on HIV patients, in which HIV enteropathy typically results in LGS. The pretreatment with *Bifidobacteria* and *Lactobacillus* attenuated the HIV enteropathy and prevented BT, but statistically significant results were observed only in the group simultaneously pretreated with probiotics and prebiotics [87]. Moreover, Villar-García et al. [88] showed that supplementation of *Saccharomyces boulardi* decreased the plasma levels of LBP in HIV individuals. Contrastingly, another study on HIV showed that nutritional mixture PMT25341 containing *Saccharomyces boulardi* was not effective in protecting patients from BT [89]. Although the results of animal models are promising, only a few studies performed on humans are available, and their results are inconclusive.

## 6. Dietary Supplements as a Potential Preventative Agents in BT

According to the recent literature, dietary supplements including amino acids, vitamins, and phytochemicals exhibit protective effects on the structure and function of the GI system. According to both in vivo and in vitro studies, these compounds can regulate microecological balance, decrease inflammation within intestines, and restore impaired barrier function by enhancing the expression of TJ proteins. Since effective treatment of LGS and BT should focus on restoring normal gut homeostasis, many authors hypothesize that this multidirectional activity of dietary supplements could possibly contribute to the treatment of patients with LGS. Nevertheless, currently, there is still a lack of evidence concerning the influence of dietary supplements on the GI system and intestinal permeability in human studies. 

### 6.1. The Role of Amino Acids in Regulation of Intestinal Barrier Integrity and Prevention of BT

Glutamine is a conditionally essential amino acid regulating many physiological processes within the organism. It is a substrate for the synthesis of many proteins, including glutathione and neurotransmitters. It also plays a crucial role in nitrogen metabolism, as it neutralizes ammonia [90,91]. Since glutamine is the main fuel for enterocytes, recent reports also highlight the importance of its function in maintaining intestinal homeostasis. Glutamine is a mitogen-activated protein kinases (MAPKs) activator that controls enterocyte proliferation and apoptosis. As an NF-κB inhibitor, glutamine also acts as an immunomodulatory agent, decreasing the production of cytokines (TNF-α or IL-6) [90]. It has been shown that enrichment of feed with glutamine significantly suppressed the expression level of an NF-κB within intestinal tissue of sepsis rats [92]. By inducing the expression of TJ proteins, glutamine also takes part in the modulation of intestinal permeability. In methotrexate (MTX)-treated Caco-2 cells, pretreatment with 10 mmol/L of glutamine prevented the decrease in expression levels of ZO-1 and occludin proteins, thus enhancing TEER value. The addition of 6-diazo-5-oxo-1-norleucine (inhibitor of glutaminase) blunted the effect of glutamine on MTX-treated cells. Interestingly, a combination of glutamine and arginine did not potentiate therapeutic effects since similar results were obtained with glutamine-only administration [93]. A small ex vivo study involving 12 patients with diarrhea-predominant IBS (IBS-D) has also evidenced that 18 h incubation with 10 mmol/L glutamine was able to restore expression of TJ proteins, in particular, claudin-1 in the colonic mucosa [94]. A randomized placebo-controlled study involving 54 patients with IBS in a controlled group showed that glutamine supplementation can also decrease intestinal hyperpermeability in humans [95]. The potential mechanism by which glutamine regulates TJ expression was investigated by Li and Neu [96]. In order to obtain glutamine deprivation, glutamine-free culture media and treatment with 4 mmol/l methionine sulfoximine (MS) (which is a glutamine synthetase inhibitor) were applied. According to the results, deprivation of glutamine significantly increased the expression of phospho/Akt protein. Additionally, the TEER value observed in the glutamine-free group was significantly decreased compared to control glutamine-treated Caco-2 cells. The effect was reversed by LY294002 and wortmannin, which are the inhibitors of phosphatidylinositol 3-kinase (PI3K). The same relationship occurred in the case of ^14^C mannitol rate and Claudin-1 expression. In both cases, treatment with PI3K inhibitors prevented intestinal hyperpermeability and suppression of Claudin-1 expression in glutamine-deprived cells [96]. This potential mechanism of action of glutamine is supported by research performed by Li et al. [97], in which glutamine also elevated TEER value in IL-13-treated Caco-2 cells via PI3K/Akt signaling pathway.

Since glutamine can affect TJ expression and thereby regulate paracellular intestinal permeability, it can be assumed that it may also prevent BT. Several studies performed in rodent models can support this hypothesis; however, data regarding the effects of glutamine in humans are still missing. Santos et al. [98] evidenced that pretreatment with 500 mg/kg/day can decrease intestinal hyperpermeability and the resulting BT in a rat model of intestinal obstruction (IO). According to the outcomes of their experiment, there were no statistically significant differences in the percentage of the dose of diethylenetriamine pentaacetate radiolabeled with technetium (^99m^Tc-DTPA) in blood between the control group and animals pre-treated with glutamine, while there was a significant increase in ^99m^Tc-DTPA blood level in mice with IO. Additionally, pre-treatment with glutamine reduces bacterial translocation to blood, liver, spleen, lungs, and MLNS compared to the IO group [98]. Moreover, glutamine is an effective factor in decreasing BT after oral administration in rats with experimentally induced sepsis [99]. 

Arginine is a conditionally essential amino acid commonly found in a wide variety of food. In the human body, arginine plays a crucial role in several physiological processes, including secretion of anabolic hormones, wound healing, and modulation of the immune response. As a precursor of nitric oxide (NO), arginine also contributes to the regulation of blood pressure [100]. Currently, there is a growing body of evidence from animal experiments supporting the beneficial effects of arginine on the GI system function [101]. Both in vivo and in vitro studies support the theory that arginine enhances intestinal barrier integrity by increasing the expression of TJ proteins such as ZO-1, claudin 1, and occludins [53,59]. Moreover, Wu et al. [102] evidenced that supplementation with arginine increases the luminal concentration of secretory (SigA) immunoglobulin responsible for defense against pathobionts and bacterial overgrowth [102]. In vivo study on the direct influence of arginine on BT was performed by Viana et al. [103]. To evaluate the therapeutic potential of arginine, animals were divided into three groups: mice fed with 2% arginine chow, mice fed with ordinary chow (intestinal damage was induced in both groups), and control animals with normal gut function. To induce intestinal barrier hyperpermeability, mice underwent a surgical operation in which the ileum was ligated. On the seventh day of the experiment, technetium radioisotope ^99m^ Tc and ^99m^ Tc marked *E. coli* were orally administered to all animals. Intestinal permeability and bacterial translocation levels were determined based on the blood and tissue radioactivity. Results showed that pretreatment with arginine significantly reduced intestinal permeability in comparison with mice fed with normal chow. Moreover, *E. coli* distribution in organs differed between groups. Arginine-treated mice displayed lower radioactivity in the liver (556.52 vs. 1154.49 cpm/g), spleen (390.00 vs. 1022.22 cpm/g), and lungs (110.52 vs. 794.12 cpm/g). More importantly, there were no statistically significant differences between arginine-treated mice and animals without intestinal barrier dysfunction [103]. A corresponding study with similar results was published by Quirinio et al. [104].

In summary, available data suggest that the use of glutamine for the prevention of BT is the most promising. However, this notion has to be confirmed by further clinical trials.

### 6.2. Effects of Vitamins on Intestinal Barrier Hyperpermeapility and BT

Vitamin D_3_ is a fat-soluble steroid mainly present in fish flesh, fish oils, egg yolk, and liver. Active form of vitamin D_3_ called calcitriol is responsible for maintaining proper bone structure and regulation of intestinal calcium, magnesium, and phosphate absorption. Synthesis of calcitriol consists of three stages: photoisomerization of 7-dehydrocholesterol to pre-vitamin D_3,_ hepatic isomerization of pre-vitamin D to cholecalciferol (vitamin D_3_), and hydroxylation of vitamin D_3_ to calcitriol in kidneys [105,106]. Besides regulating calcium and phosphate metabolism, calcitriol is evidenced to modulate immune system response and influence intestinal homeostasis. Vitamin D deficiency is commonly diagnosed in IBD patients, and it is one of the risk factors associated with the occurrence of CD [107]. The effect of vitamin D on intestinal epithelium was investigated in the LPS-treated IEC-18 cell line by Lee et al. [108]. To assess the potential therapeutic action of tested compound intestinal cell monolayer permeability (measured by fluorescein Isothiocyanate-dextran (FD4) assay), cell viability (measured by MTT test) and expression levels of TNF-**α** and IL-6 (measured by real-time PCR) were examined after 3 days of incubation with vitamin D. Consistent with the results, treatment with vitamin D (10 μM) significantly reduced intestinal permeability (19.11 ± 1.9 vs. 32.56 ± 2.0) compared to LPS-treated cells. Moreover, there were no differences between the tested and control group (19.11 ± 1.9 vs. 20.89 ± 1.9). Of note, a decrease in expression levels of TNF-**α** and IL-6 may indicate the involvement of vitamin D in regulating the immune response and inflammation [108]. Although vitamin D decreases intestinal permeability, in vitro study performed on SKCO15 cell line (commonly used in research concerning TJs) by Zhang et al. [70] evidenced that vitamin D enhances expression of pore-forming protein claudin-2 in a dose-dependent manner. Of note, vitamin D receptor (VDR) KO mice display decreased claudin-2 mRNA and claudin-2 protein levels compared to WT. CHIP assay verifying the mechanism of action of vitamin D evidenced that VDR binds to claudin-2 promotor and acts as a transcriptional factor regulating the expression of this particular TJs protein [109]. However, contradictory results were obtained by Stio et al. [110], who showed that expression of the claudin-2 protein was higher in biopsies from inflamed parts of the colon and the rectum in patients with active UC compared to non-inflamed tissue. Incubation with 100 nM 1,25(OH)_2_D_3_ resulted in the decrease in pore-forming claudin-2 and cytokines, namely IL-13 and IL-6 [110]. As Zhu et al. [111] reported, vitamin D can affect intestinal homeostasis by directly influencing the gut microbiome [111]. Prevention of vitamin D conversion into its active form (1,25(OH)_2_D_3_) achieved by knocking out *CYP2B1* gene resulted in intestinal dysbiosis followed by other functional changes. CYP2B1 KO mice were characterized by increased abundance of *Akkermansia muciniphila* and *Solitalea canadensis* and decreased abundance of *Bacterioides uniformis*, *Bacterioides eggerthii*, and *Roseburia inulinivorans*. Since *A. muciniphila* is a mucin degrading bacterium and *B. uniformis* plays an essential role in SCFAs production, CYP2B1 KO mice displayed respectively lower mucus thickness and decreased feces concentration of butyrate (approximately 8–9 mM/L vs. 20–21 mM/L) compared to WT. Additionally, 1,25(OH)_2_D_3_ deficiency resulted in increased BT to the MLNs [111]. Changes in the intestinal microbiome induced by vitamin D are presumably modulated by VDR. According to Jin et al., VDR KO are characterized by elevated abundance of *Clostridia* spp. and *Bacteroides* spp. and reduction of *Lactobacillus* spp. in fecal microbiome compared to WT. Simultaneously, there are no major changes in the cecal bacteria composition in VDR KO mice when compared to WT [112]. Despite a growing number of both in vitro and in vivo evidence demonstrating beneficial effects of vitamin D on intestinal permeability, there is still a lack of data concerning the influence of this vitamin on human intestinal homeostasis. One of a few studies investigating the association between vitamin D serum levels and intestinal permeability was performed by Eslamian et al. [113]. Researchers evidenced that deficiencies of vitamin D (serum level below 20 ng/dL) correlated with increased zonulin concentration and plasma endotoxin serum level in critically ill patients [113]. A study concerning the direct influence of vitamin D on BT in cirrhotic rats was performed by Lee et al. [114]. According to the outcomes of the experiment, administration of vitamin D_3_ at the dose of 0.1 µm/kg/day reduced BT to MLNs. This effect was associated with upregulation of occludin-1 and claudin expression levels, respectively, in the small intestine and within the colon. Alvarez et al. [115] also showed that sufficient (>75 nmol/L) plasma levels of 25(OH)D in HIV/hepatitis C virus (HCV) co-infected patients correlated with lower BT (measured by bacterial DNA copies in plasma) compared to subjects with insufficiencies of vitamin D.

Supplementation with vitamin A is another therapeutic strategy postulated for the treatment of LGS. Vitamin A is a group of fat-soluble steroids (retinol, retinal and retinoic acid) found in butter, dairy, egg yolk, and vegetables, including carrots, pepper, spinach, and sweet potatoes [116]. In the human body, vitamin A mainly promotes good eyesight, but novel studies indicate its role in immune response, wound healing, and even cancer development [117,118]. There is also a growing amount of evidence demonstrating the important influence of vitamin A on gut homeostasis. Vitamin A-deficient mice have significantly lower levels of *Bacterioidates* phyla and higher levels of *Staphylococcaceae* family members. In contrast, the enhanced abundance of butyrate-producing bacteria-*Clostridium ramosum* in vitamin A sufficient mice results in increased concentration of SCFAs, which display protective effects on intestinal barrier integrity [119]. In addition, vitamin A-deficient rats are characterized by increased translocation of *E. coli* into MLNs and kidneys [120]. In addition to the influence on intestinal bacteria composition, vitamin A exhibits immunomodulatory properties. Retinoic acid enhances T-cell proliferation, increases the concentration of IL-10 and IL-22, and decreases levels of INF-γ and IL-17 [121]. As Okayasu et al. [122] reported, supplementation with vitamin A inhibits the development of dextran sodium sulfate (DSS)-induced colitis and colon cancer in mice as demonstrated by reduced macro- and microscopic scores [122]. According to He et al. [123], vitamin A also has a direct impact on intestinal barrier permeability [123]. In LPS-treated IPEC-J2 cells, treatment with 0.1 μM of vitamin A increased TEER value compared to control. This effect was mediated via increased expression of TJ proteins, including ZO-1, Occludin, and Claudin-1 [123]. 

Given the scarcity of clinical trials assessing the effect of vitamins on the intestinal barrier function, a firm conclusion cannot be drawn. However, in the authors’ opinion, the abundance of promising in vitro and in vivo studies warrants further clinical evaluation of this strategy. 

### 6.3. Influence of Plant-Based Dietary Supplements on Intestinal Barrier

Phytochemicals are chemical compounds derived from plants. From a chemical perspective, phytochemicals can be divided into polyphenols, alkaloids, and terpenoids. Although phytochemicals do not play a key role in maintaining proper body function, their antioxidant, vasoprotective, antiseptic, analgesic, immunomodulatory, neuroprotective, and hepatoprotective activities have been used for centuries in medical practice. With the growing development of synthetic and, more recently, biological drugs, phytochemicals have lost their therapeutic significance. Despite the fact that plant-derived drugs cannot be used in severe health problems such as cancer, mental disorders, hypertension, or metabolic disorders, they can contribute to patient treatment as a complementary therapy [124]. Recent in vivo and in vitro studies on the influence of phytochemicals (mostly polyphenols) on the GI system have revealed that these compounds participate in intestinal permeability regulation. According to Nunes et al. [125], a polyphenolic extract derived from red wine (RWE) decreases intestinal permeability via upregulation of TJ proteins in cytokines-treated Caco-2 cells [125]. Moreover, RWE enhanced occludin, claudin-5, and ZO-1 expression in a dose-dependent manner [125]. Beneficial effects on the intestinal barrier of other phytochemicals are presented in Table 1. Some literature data also suggest that phytochemicals can prevent and decrease BT in rodents. In the mouse model of DSS-induced colitis, administration of fermented barley and soybean mixture (BS) containing isoflavonoids resulted in increased expression of TJ proteins (claudin-1 and Occludin) followed by a decrease in BT to MLNs. Additionally, compared to the DSS group, treatment with 200 mg/kg of BS significantly enhanced the abundance of *Lactobacilli* spp. in the colon [126]. Additionally, a decrease in BT to liver, spleen, and MLNs was seen after administration of 5 mg/kg/day of quercitrin in the same animal model of intestinal inflammation. This effect was possibly due to the anti-inflammatory properties of the investigated compound, as in the experiment, researchers observed a decrease in TNF-α serum concentration and MPO activity [127]. Many phytochemicals also exert antioxidant activity, which reduces oxygen-free radical levels and prevents oxidative cellular damage. Thymoquinone- bioactive component of *Nigella sativa* L. (family Ranunculace) is evidenced to decrease total oxidant activity (AOA) and regulate intestinal inflammation by suppressing the expression of proinflammatory cytokines (IL-β and IL-8). In rats with intestinal obstruction, thymoquinone prevented intestinal damage as showed by inflammation scores of ileum, liver, and MLN and decreased biochemical parameters such as TNF-α, IL-6, IL-1β, and CRP level. These changes were followed by a reduction of BT to liver, spleen, MLN, and blood when compared to the IO group [128]. Even though there are many studies on cell lines and animal models confirming the positive effect of dietary supplements on the permeability of the intestinal barrier and BT, there are still too few reports on the effect of these compounds in humans to draw clinically relevant conclusions.

## 7. Drugs Proposed in Prevention of BT

### 7.1. Current Treatment and Prophylaxis of BT

Selective digestive decontamination (SDD) is a strategy used in treatment and prevention in patients with the risk of developing BT. SDD refers to the eradication of Gram-negative bacteria, especially *Enterobacter* spp. and fungi, by low doses of antibiotics. Elimination of potentially pathogenic strains can lower the probability of dysregulation of the immune system due to infection and promote the growth of Gram-positive bacteria, including *Lactobacillus* spp., which protects intestinal barrier integrity [129]. Effectivity of SDD is well confirmed by animals and human studies. Administration of tobramycin in conjunction with polymyxin E significantly decreases the incidence of BT to MLNs in mice feed by total parenteral nutrition (TPN) and the number of Gram-negative bacteria in caecum compared to animals fed with only TPN. In addition, SDD reverses to a standard concentration of intestine IgA in ORAL-TPN treated mice [130]. The most effective drug used in SDD is norfloxacin, which is recommended by the American Association for Study of Liver Diseases (AASLD) and European Association for the Study of Liver (EASL) in the prevention of SBP, which is supposed to be a consequence of BT to MLNs in patients with cirrhosis, low-proteins ascites, and renal or liver failure [131]. According to a randomized placebo-controlled study performed by Fernardez et al. [132], all one-year prophylaxis with 400 mg/day of norfloxacin in patients with cirrhosis and ascites significantly reduced the 1-year probability of developing SBP (41% vs. 61% in the placebo group). Additionally, long-term administration of norfloxacin increased the 3-month and 1-year probability of survival. In addition to reducing the overgrowth of Gram-negative bacteria, norfloxacin normalizes serum LBP and endotoxin level and decreases the concentration of proinflammatory cytokines, including TNF-α, IL-6, and NOx/creatine ratio in cirrhotic patients with high LBP plasma levels. However, these effects were only observed in patients with high LBP serum levels, and none of the above-mentioned changes were detected in patients with normal LPB plasma levels, suggesting that the potential mechanism of action is not based on the anti-inflammatory properties of norfloxacin [133]. Although prevention of BT by the use of antibiotics is an effective strategy, due to antibiotic resistance and potential disruption in gut microbiome composition, it should be used only in patients with severe diseases such as cirrhosis with a high risk of developing infections in non-GI tissues caused by BT [131]. Evidence for BT in various GI disorders are presented in Table 2.

### 7.2. Potential Use of Anti-Inflammatory Drugs in Prevention of BT

Although diet, dietary supplements, and probiotics can improve intestinal barrier integrity and therefore prevent BT, they may not be a sufficient therapeutic strategy in patients with more severe forms of LGS. In addition to diet, many authors mention anti-inflammatory drugs as potential therapeutic agents in the treatment of BT, and since inflammation can disrupt the intestinal homeostasis and induce intestinal hyperpermeability due to a decrease in TJ proteins expression and intestinal damage, this strategy may play a key role in patients with LGS-like symptoms. Although data from in vivo and in vitro studies are promising, there is still a lack of evidence confirming the effectiveness of anti-inflammatory drugs in human studies [129]. Humanized anti-TNF-α monoclonal antibodies (anti-TNF-α mAB) such as infliximab are used in the treatment of rheumatic and autoimmune diseases, including rheumatoid arthritis or psoriasis as well as GI diseases such as CD and UC. Besides immunomodulatory properties, anti-TNF-α mAb has been shown to affect intestinal permeability and BT [144,145]. Treatment with 5 mg/kg body weight of infliximab in patients with long-standing active CD results in a significant decrease in lactulose/mannitol (L/M) rate after seven days of drug injection. Moreover, there were no statistically significant differences in the L/M rate between the healthy control group and patients receiving investigated antibodies [144]. As Frances et al. [145] evidenced, anti-TNF-α-mAb also decreases BT in a rat model of CCl_4_-induced cirrhosis. The detailed mechanisms by which anti-TNF-α mAb mitigate intestinal hyperpermeability still remain unknown; however, it probably abolishes down-regulation of expression levels of crucial TJ proteins (occludin, claudin-3, claudin-5, and claudin-8) caused by increased concentration of TNF-α within GI tissues [144].

Pentoxifylline (PTX) is a xanthine derivative indicated for the treatment of circulatory disorders caused by diabetes or atherosclerosis. In addition to its anti-aggregation activity, several reports also demonstrate that PTX has immunomodulatory properties and can increase the survival rate in rats with sepsis. As a phosphodiesterase inhibitor, PTX inhibits TNF-α and NF-κB synthesis. Kocdor et al. [146] investigated the influence of PTX on BT in a rat model of intestinal impairment caused by intestinal obstruction. According to the outcomes of the experiment, specimens collected from MLNs, 12 h after PTX administration (in a dose of 50 mg/kg), were characterized by a significant decrease in bacteria (*E. coli* and *S. aureus*) level compared to rats treated by placebo. In addition, specimens collected from liver and blood samples did not show the presence of any bacteria. Twenty-four hours after administration of the first dose of PTX (the second dose was administered after 12 h), BT to MLNs was observed in only 40% compared to 90% in the control group [146]. A similar study was performed by Köylüoğlu et al. [147]; however, in this experiment, animals were subjected to hemorrhagic shock. Treatment with 25 mg/kg of PTX completely prevented bacteria translocation to MLNs, spleen, liver, and blood. In the control group, BT to MLNs occurred in four rats (out of eleven), to spleen and blood, respectively, in one rat, and to liver, in two rats. *E. coli* was the most common bacteria detected within specimens, but *Staphylococcus*, *Streptococcus*, and *Enterococcus* were also isolated. Although in vivo studies concerning the influence of PTX on bacteria translocation are promising, there is still a lack of data regarding the use of this drug in humans [147].

### 7.3. Prokinetics and Laxatives Influence Intestinal Permeability and BT

Prolonged intestinal transit due to intestinal hypomotility can promote the overgrowth of pathogenic bacteria and their adhesion to the intestinal wall, which both can contribute to BT. Low bowel motility and SIBO are observed in patients with liver cirrhosis who also have a higher risk of development of SBP [134,135,136]. In detail, orocecal transit time (OCT) is significantly prolonged in cirrhotic patients with intestinal bacteria overgrowth (IBO) as compared to patients without IBO [137]. Therefore, some authors propose a therapeutic strategy concentrated on increasing intestinal motility by prokinetics, among which the best studied is cisapride.

Cisapride is a 5-HT_4_ agonist with 5-HT_3_ antagonist activity exerting its effect by releasing acetylcholine from postsynaptic neurons in the enteric nervous system. It stimulates saliva secretion, accelerates gastric and cecal emptying, and shortens transit time [138]. In cirrhotic rats, intragastric administration of cisapride reduced BT to MLNS, liver, and spleen and decreased endotoxin level and IBO compared to the placebo group. Additionally, a decrease in intestinal permeability (measured by 24-hour urinary excretion of ^99m^Tc-DTPA) was observed [139]. Likewise, Wang et al. [140] reported that improvement of intestinal transit with cisapride prevented *E. coli* overgrowth and significantly decreased BT in rats with acute liver failure. In human studies, administration of norfloxacin in conjunction with cisapride in patients with cirrhotic ascites reduced the probability of SBP at 12 months to 21.7% compared to 56.7% in the norfloxacin group. Additionally, the actuarial probability of death at 18 months was lower in the norfloxacin+cisapride group (6.2%) compared to the norfloxacin alone (20.6%) [141]. According to a small study performed by Pardo et al. [137] treatment with cisapride completely abolished IBO caused by Gram-negative bacteria *(E. coli*, *Enterobacter*, *Klebsiella* sp., and *Pseudomoma* sp.) in four out of five cirrhotic patients included in the experiment. Despite its effectiveness, treatment with cisapride carries a risk of development of serious and potentially fatal arrythmia and has therefore been withdrawn from the market in 2000 [138,142]. A drug with an analogous mechanism of action but devoid of arrhythmogenic properties is mosapride [143]. Similarly to cisapride, mosapride reduced endotoxemia and BT in cirrhotic rats in the experiment carried out by Xu et al. [143]. However, this is the only in vivo study investigating the influence of mosapride administration on intestinal microbiota composition and BT; therefore, more data are needed to evaluate potential effectivity and possibly transfer the outcomes to human studies.

Lubiprostone is an oral chloride channels-2 activator indicated for the treatment of chronic idiopathic constipation and IBD with constipation in adults [148]. In addition to laxative properties, Lubiprostone is the first antidiarrheal drug with evidenced activity decreasing intestinal barrier hyperpermeability. According to Nishii et al. [149], pretreatment with Lubiprostone increased TEER in a dose-dependent manner in INF-γ treated Caco-2 cells [149]. Zong et al. [150] discovered that a decrease in intestinal permeability in cortisol-treated Caco-2 cells was modulated by enhancement of occludin and claudin-1 expression [150]. Moreover, administration of Lubiprostone to rats stressed by water avoidance improved their body weight and prevented visceral hyperalgesia [150]. In addition to regulatory effects on TJ proteins, Lubiprostone exhibits anti-inflammatory properties. In rats with indomethacin-induced enteropathy, administration of Lubiprostone (0.1 mg/kg) suppressed myeloperoxidase (MPO) activity, reduced TNF-α expression, and prevented bacterial invasion in small intestines [151]. The first prospective randomized study concerning the influence of Lubiprostone on gut permeability in humans was performed by Kato et al. [3]. The control group consisted of patients in whom intestinal hyperpermeability was induced by 7 days of administration of diclofenac at a dose of 75 mg/day. Treatment with Lubiprostone (24 µg/day) in the study group started on the 7th day and lasted until the 28th day of the experiment. Intestinal barrier permeability was assessed based on the lactulose–mannitol ratio (LMR) in urine and level of endotoxemia. According to results, on the 28th day, the LMR ratio was significantly lower in Lubiprostone-treated patients compared to control. Since one month’s administration of NASIDs may not induce severe intestinal hyperpermeability, there were no statistically significant differences in LPS blood concentration between the study and the control group [3]. As in the case of diet, probiotics, and dietary supplements, there is still a lack of data confirming the effectiveness of anti-inflammatory, prokinetic, or laxative drugs in the prophylaxis of BT. Given that SDD is currently the only recommended way to prevent BT in humans, which can be associated with serious side effects such as the development of antibiotic resistance, more research should focus on finding other potential therapeutic solutions.

### 7.4. Growth Hormone Prevents BT in Animal Models

Growth hormone (GH) is a peptide hormone secreted by the pituitary gland, which stimulates proliferation, growth, and tissue repair. Since growth hormone receptors (GHR) are widely expressed by various cells along the entire length of the GI tract, the influence of GH on gut homeostasis has been a subject of much research. Intestine-specific GH receptor KO (IntGHRKO) mice display gender-specific changes. Male IntGHRKO mice exhibit a significant increase in occludin expression, decrease in fat absorption and decrease in large intestinal length, whereas female IntGHRKO is glucose intolerant [152]. GHRs are also expressed by immune cells, with the highest expression level in B cells and the lowest in T cells [153]. GH increases serum and intestinal fluid secretory IgA level, which prevents bacterial adhesion to the mucosal surface and therefore limits the number of bacteria within the GI tract [154]. A recent study performed by Jensen et al. [155] also indicates that GH may also be involved in the regulation of the intestinal microbiome composition [155]. GH gene-disrupted mice display shifts in the *Firmicutes*/*Bacteroides* ratio and a decrease in the number of *Proteobacteria* and *Campylobacteria* compared to WT animals [155].

A series of studies evaluated the effect of GH on BT in animals models [154,156,157,158]. Pre-treatment with GH in irradiated rats prevented weight loss and decreased mortality and BT to MLNs from 69% to 29% [156]. Likewise, a significant decrease in BT to MLNs, liver, and blood was reported by Kymakci et al. [154] in rats with partial intestinal obstruction (PIO). This effect was probably partially modulated by enhancement in serum and intestinal fluid IgA level and restoration of physical barrier integrity, as administration of GH increased the number of small intestinal goblet cells and length of mucosal villi [154]. Contraindicatory to Kymakci et al. [157] experiment, reduction of intestinal epithelium cells apoptosis, intestinal permeability, and BT after subcutaneous injections of recombinant human growth hormone (rhGH) in LPS-treated rats did not correlate with the decrease in mortality. Some authors also propose the use of GH in conjunction with GLN to achieve better therapeutic effects [158,159]. According to Jung et al. [158], a combination of GH and GLN reduced BT to portal blood in sepsis rats; however, there were no statistically significant differences between GH- and GLN-treated animals. Nevertheless, treatment with GH and GLN significantly decreased bacterial colony counts obtained from MLNs cultures and enhanced mucosal thickness when compared to GH or GLN alone [158]. In humans, the beneficial effects of GH and GLN were shown by Tang et al. [159]. In a prospective, randomized, and controlled clinical study, TPN supplemented with rhGH and GLN decreased postoperative blood concentration of TNF-α and CRP and enhanced the level of IgG, the ratio of CD4/CD8, and the number of CD4 cells compared to control in cirrhotic patients after portal hypertension surgery. Additionally, in a supplemental group, the increase in villus height and crypt depth was noted. These changes were also accompanied by a reduction of intestinal permeability, as demonstrated by the decrease in the L/M ratio [159]. Interestingly, in a corresponding study performed among patients after abdominal surgery, subcutaneous administration of GH did not produce any changes with statistical significance [160]. These results indicate that treatment with GH needs further validation, and perhaps its efficacy depends on the type of performed surgery.

### 7.5. Gastric Acid Protects from SIBO and BT

Current literature provides a growing number of evidence from both animal and human studies that gastric acid influences the composition of the intestinal microbiome. Inhibition of gastric acid secretion by omeprazole predispose to duodenal bacterial overgrowth, which might potentially favor BT [161,162,163,164]. Similar changes were also observed during H2-antagonists therapy such as cimetidine; however, the effects were weaker [165]. Treatment with proton pump inhibitors (PPI) is also associated with shifts within the intestinal microbiome, especially an increase in the abundance of *Enterococcus*, oral *Streptococcus*, *Staphylococcus* genera, and potentially pathogenic *Escherichia coli* [166]. There is still a lack of data to support the impact of increased gastric pH level on BT; however, Choi et al. [167] and Bajaj et al. [168] evidenced that treatment with PPI inhibitors correlated with increased risk of SBP in cirrhotic patients with ascites [167,168]. Replacing PPI with H2 antagonists or inclusion of drugs protecting intestinal mucosa without increasing pH such as prostaglandin E (PGE) may form a new treatment plan for patients with hyperchlorhydria who are also at higher risk of developing bacterial infections due to BT. Interestingly, misoprostol, a PGE 1 analog can prevent BT in animal models [169,170,171]. According to Gurleyik et al. [170], intragastric administration of 40 mg of misoprostol to jaundiced rats resulted in decreased mucosal damage score, increased mucosa thickness, and, more importantly, decreased BT. The incidence of BT to MLNs and blood reduced from 90% to 60% and from 30% to 10%, respectively, in the misoprostol-treated group.

## 8. Conclusions and Future Perspective

Translocation of bacteria or endotoxin in LGS is associated with GI diseases. Elevated intestinal barrier permeability may be the first step in the development of various GI disorders, given that undigested food particles, bacterial toxins, and germs can pass through the “leaky” gut wall and into the bloodstream, triggering the immune system and causing persistent inflammation. Of note, clinicians have a large problem with recognition of LGS because there are several symptoms overlapping with other common GI diseases, such as diarrhea or constipation. Moreover, there is no gold standard procedure for a clear characterization of the barrier function. Currently available tests for barrier function measure very different endpoints, and therefore, their clinical significance and relevance are unclear [10].

Intestinal bacteria play an important role in the human body. Most of the bacteria have a positive effect on our health and contribute to many natural processes, including metabolism of indigestible compounds, synthesis of vitamins, protection against pathogen colonization, and contribution to the host immune system. On the other hand, the imbalanced gut bacterial composition leads to serious health problems. Various factors may influence BT in the GI tract, e.g., (1) diet, mainly fat and saccharide composition, (2) probiotics, (3) dietary supplements, such as amino acids and vitamins, and (4) medicines. Many reports indicate that high-fat and high-sugar diets disturb the composition of the bacterial microflora and induce a harmful effect on intestinal permeability. Importantly, probiotics mainly consist of *Bifidobacteria* and *Lactobacillus* to prevent colon endotoxemia. Furthermore, vitamins, especially vitamins D and A, also play an important role in the integrity of intestinal epithelium. Patients suffering from IBD have decreased levels of vitamin D_3_, and it is also associated with a higher risk of *Clostridium difficile* infection in IBD individuals. Additionally, lower vitamin D concentration correlates with an increase in zonulin and plasma endotoxin serum level, suggesting their further impact on intestinal permeability. Available treatments for patients with LGS are based on the underlying condition, which often includes leaky gut as a symptom (Figure 2). Accordingly, anti-inflammatory drugs and immune system suppressors are prescribed in patients with IBD, whereas anticholinergics and ligands of serotonin receptors are recommended for patients with IBS, etc. Odenwald et al. [172] noticed that no FDA-approved agents targeting epithelial barriers are presently available. However, promising approaches to target the LGS are being investigated. Interestingly, the effect of nutrients, probiotics, and even vitamins on barrier function and the alleviation of clinical symptoms of GI diseases though preventing BT is impressive, suggesting that these ingredients should be considered as promising candidates for the treatment of patients suffering from the LGS.

## Figures and Tables

**Figure 1 ijms-23-03204-f001:**
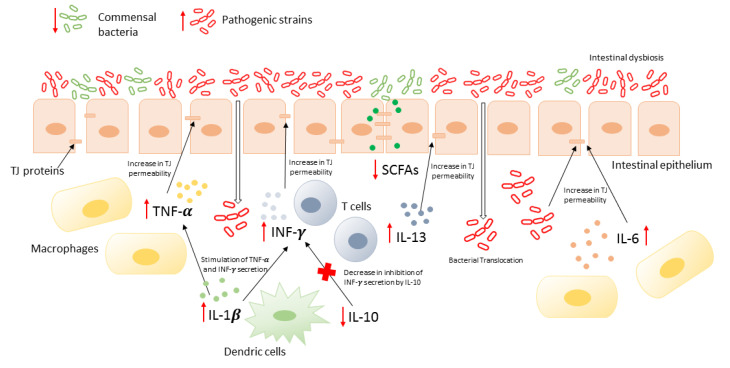
Consequences of bacterial translocation.

**Figure 2 ijms-23-03204-f002:**
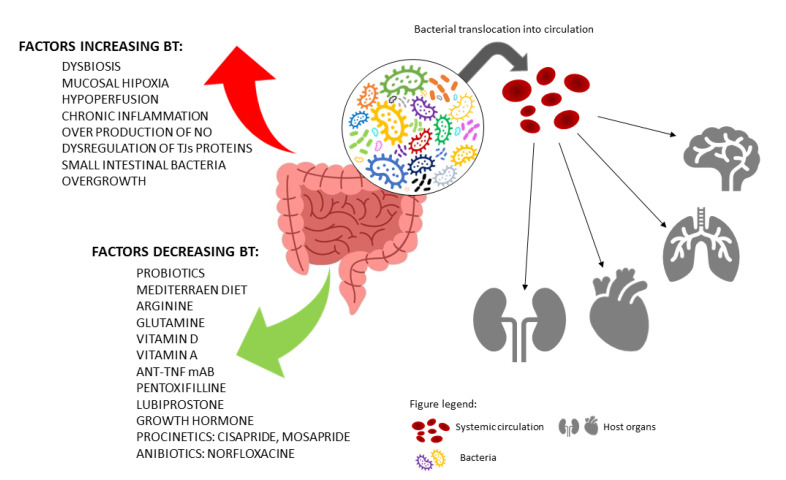
Factors influencing bacterial translocation (made with the use of free materials from www.canva.com accessed on 15 October 2021).

**Table 1 ijms-23-03204-t001:** Effect of selected phytochemicals on intestinal barrier integrity.

Plant Source of Compound	Active Compound	Experimental Model	Effect on Intestinal Barrier Integrity	References
*Berberis aristate* (Indian Barbery) *Hydrastis canadensis* (Goldenseal)	Berberine	Mouse model of IBS-D	Decreased intestinal permeability via upregulation of TJ proteins (ZO-1, claudin-1) expression. Reduced expression of TNF-α.	Hou et al. [88]
*Sumbucus nigra* (Elderberry) *Vaccinium myrtillus* (European blueberry) *Vitis vinifera* (Common grape vine)	Anthocyanis (cyjanidin and delphinidin)	Mouse model of HFD-associated increased intestinal permeability	Decreased intestinal permeability via upregulation of TJ proteins (occludin, ZO-1, and claudin-1) expression. Decreased expression of NADPH oxidase (NOX1 and NOX4). Reconstruction of physiological microbiota composition and decreased level of endotoxemia.	Cremonini et al. [89]
*Curcuma longa* (Turmeric)	Curcumin	LPS-treated Caco-2 and HT-29 cells	Decreased secretion of pro-inflammatory cytokine IL-1β and increased secretion of anti-inflammatory cytokine IL-10. Decreased expression of MLCK. Restoration of proper TJ organization.	Wang et al. [90]
*Reynoutria japonica* (Japanese knotweed)	Resveratrol	H_2_O_2_-treated IPEC-J2 cells	Increased expression of TJ proteins (ZO-1, occludins, and claudin-1). Increased cell viability and decreased apoptotic rate.	Zhuang et al. [91]
*Scutellaria baicalensis* (*Baikal skullcap*) *Scutellaria lateriflora* (American skullcap)	Baicalin	LPS-treated IEC-6 cells	Decreased concentration of TNF-α and IL-6. Increased expression of claudin-3, occludin, and ZO-1.	Chen et al. [92]
*Rheum palmatum* (Chinese rhubarb)	Rhein	TNF-α-treated IEC-6 and LPS-treated IEC-6	Increased expression of ZO-1. Decreased expression of pro-inflammatory cytokines: IL-1β and IL-6. Decreased intestinal permeability measured by TEER.	Zhuang et al. [93]

**Abbreviations:** HFD—high-fat diet; IBS-D—diarrhea predominant irritable bowel syndrome; IL-1β—interleukin 1β; IL-6—interleukin 6; IL-10—interleukin 10; LPS—lipopolysaccharide; MLCK—myosin light-chain kinase; NOX1—NADPH oxidase 1; NOX4—NADPH oxidase 4; TEER—transepithelial resistance; TJ—tight junctions; TNF—α-tumor necrosis factor α; ZO-1—zonula occludens-1.

**Table 2 ijms-23-03204-t002:** Evidence of bacterial translocation in various alimentary tract disorders.

Condition	Method of Detecting Bacterial Translocation	Incidence of Bacterial Translocation among Patients (in %)	References
Ulcerative colitis	Presence of BactDNA in serum	51.7%	Guti-acerrez et al. [134]
	Presence of 16S ribosomal RNA gene Segments in intestinal lymph follicles	40%	Chiba et al. [135]
Crohn’s disease	Presence of BactDNA in serum	42.4%	Guti-acerrez et al. [134]
	Presence of bacteria in MLNs	33%	Ambrose et al. [136]
	Presence of BactDNA in blood	34%	Gutierrez et al. [137]
	Presence of 16S ribosomal RNA gene Segments in intestinal lymph follicles	28%	Chiba et al. [135]
Acute pancreatitis	Presence of BactDNA in blood	19.3%	De Madaria et al. [138]
	Presence of BactDNA in blood and ascitic fluid	32.1%	Such et al. [139]
	Presence of BactDNA in blood and ascitic fluid	41.2%	Frances et al. [140]
Cirrhosis	Presence of BactDNA in blood	33%	Gimenez et al. [141]
	Presence of bacteria in MLNS	In Child-Pugh A patients: 3.4% In Child-Pugh B patients: 8.1% In Child-Pugh C patients: 30.8% (Child-Pugh scale is used to determine the prognosis of patients with conditions leading to hepatic failure. It considers bilirubin level, albumin level, prothrombin time, ascites, and hepatic encephalopathy)	Cirera et al. [142]
	Presence of BactDNA in blood	31.8% in patients with SIBO 4.8% in patients without SIBO	Jun et al. [38]
Diabetes-2	Presence of bacteria DNA in blood	28%	Sato et al. [143]

**Abbreviations:** BactDNA—bacterial DNA; MLNS—mesenteric lymph nodes; SIBO—small intestinal bacterial overgrowth.

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
