# Peer review of "Preventing Bacterial Translocation in Patients with Leaky Gut Syndrome: Nutrition and Pharmacological Treatment Options"

_ijms, 2022, doi:10.3390/ijms23063204_

Round 1

Reviewer 1 Report

This review provides a comprehensive overview about the different options of prevention and treatment of bacterial translocation in leaky-gut syndrome (LGS).

Minor comments:

  • a lot of typos/spelling errors (e.g. "is showed"....) are in text, please correct by native speaker
  • give full meaning of all abbreviations when first mentioned
  • provide a complete list of abbreviations
  • explain throughout the manuscript and abstract: "outside the gastrointestinal tract", "outside GI", and "bacterial translocation", i.e. you also mean that bacteria and other pathogens transfer "into the body/into the system"
  • state which time period is spanned by your literature search 
  • line 56: "preventing BT ...therapeutic opportunity" = paradox
  • Figure 1: describe in figure legend what is drawn, and label arrows and molecules in the picture; present in clear colours or grey scale (e.g. commensals and pathogens should be better distinguished)
  • line 79: give short results sentence: how many papers were found, how many included, how long was search time frame
  • line 113. "mucin-degrading species" - is this a positive or negative effect?
  • line 115: IPEC-J2 cells: define/describe
  • line 126: discuss whether BT-association with different diseases like CD, UC.....is the reason or the result; so is BT the reason for the diseases, or is the disease first and BT thereafter (might be different for CD vs. abdominal surgery)
  • lines 151-155: role of microbiota/bacteria unclear, just IL-10-deficiency discussed
  • line 193: define resistin-like
  • line 233: what are the control groups in this paper?
  • figure 2: label arrows, describe drawing in figure legend; typo: pentoxifyllin
  • Table 1: give English names for plant sources in addition to botanical names
  • line 570: describe the mode of action of pentoxifyllin
  • Table 2: sort clearly / provide lines between paragraphs / percentages
  • Tabel 2: describe "child-pugh"
  • line 696: should be PPI
  • line 707: rephrase to "...translocation of bacteria or endotoxin in LGS is associated with GI diseases."
  • line 729: what is meant by increased zonulin? increased plasma levels of zonulin and endotoxin?
  • please give a short summary sentence below all your individual paragraphs (i.e. after diet, probiotics, supplements...) which conclusions can be drawn or what research is missing
  • line 739: summing up all evidence, can we really conclude on treatment of LGS, or is it rather prevention?

Author Response

Please find our responses to Reviewers comments in the attached file.

Reviewer 2 Report

The review article “Preventing bacterial translocation in patients with leaky gut syndrome: nutrition and pharmacological treatment options” is interesting and written well. The manuscript covers a different aspect of dietary intake on gastrointestinal bacterial translocation. The manuscript looks sound and broadly discussed.

Author Response

(The authors gave the same response as above.)
